# Assembling Near-Infrared Dye on the Surface of Near-Infrared Silica-Coated Copper Sulphide Plasmonic Nanoparticles

**DOI:** 10.3390/nano13030510

**Published:** 2023-01-27

**Authors:** Oleg Dimitriev, Yuri Slominskii, Mariangela Giancaspro, Federica Rizzi, Nicoletta Depalo, Elisabetta Fanizza, Tsukasa Yoshida

**Affiliations:** 1V. Lashkaryov Institute of Semiconductor Physics NAS of Ukraine, pr. Nauki 41, 03028 Kyiv, Ukraine; 2Graduate School of Organic Materials Science, Yamagata University, Jonan 4-3-16, Yonezawa 992-8510, Japan; 3Institute of Organic Chemistry NAS of Ukraine, 5 Murmanska Str., 02660 Kyiv, Ukraine; 4Chemistry Department, University of Bari, via Orabona 4, 70125 Bari, Italy; 5CNR-Institute for Chemical and Physical Process, SS Bari, via Orabona 4, 70125 Bari, Italy

**Keywords:** plasmonic nanoparticles, near-infrared dye, silica-coated copper sulphide, surface chemistry, dye immobilization

## Abstract

Functionalization of colloidal nanoparticles with organic dyes, which absorb photons in complementary spectral ranges, brings a synergistic effect for harvesting additional light energy. Here, we show functionalization of near-infrared (NIR) plasmonic nanoparticles (NPs) of bare and amino-group functionalized mesoporous silica-coated copper sulphide (Cu_2-x_S@MSS and Cu_2-x_S@MSS-NH_2_) with specific tricarbocyanine NIR dye possessing sulfonate end groups. The role of specific surface chemistry in dye assembling on the surface of NPs is demonstrated, depending on the organic polar liquids or water used as a dispersant solvent. It is shown that dye binding to the NP surfaces occurs with different efficiency, but mostly in the monomer form in polar organic solvents. Conversely, the aqueous medium leads to different scenarios according to the NP surface chemistry. Predominant formation of the disordered dye monomers occurs on the bare surface of mesoporous silica shell (MSS), whereas the amino-group functionalized MSS accepts dye predominantly in the form of dimers. It is found that the dye–NP interaction overcomes the dye–dye interaction, leading to disruption of dye J-aggregates in the presence of the NPs. The different organization of the dye molecules on the surface of silica-coated copper sulphide NPs provides tuning of their specific functional properties, such as hot-band absorption and photoluminescence.

## 1. Introduction

The functionalization of colloidal nanoparticles (NPs) with organic dyes has attracted significant interests due to the promising applications of these hybrid systems in different fields, such as photovoltaics [1,2,3,4], photoswitchable non-fluorescent thermochromic hybrid probes [5], heterogeneous photocatalysis [6], upconversion and down-conversion of energy [7,8,9,10], sensing and fluorescence imaging [11,12,13,14,15,16,17], binding and removal of toxic dyes and heavy metal ions from water [18,19,20], surface-enhanced Raman spectroscopy [21,22], etc. Depending on the surface chemistry of NPs, binding and immobilization of dye molecules on the surface of NPs can occur due to covalent bonding, H-bonding and/or physisorption. Up to now, NPs of different natures, such as inorganic oxides [23], metals [24,25], chalcogenides [26], carbon quantum dots [27], magnetic NPs [28], etc., were exploited as a host decorated with guest dye molecules.

It is worth noting that dye aggregation and assembling into ordered structures in solution is driven not only by various dye–dye intermolecular interactions, including van der Waals force, hydrogen bonding, hydrophobic interaction, etc. [29], which is related to the dye chemical structure characteristics, and by the dye–solvent interactions, which depend on the dispersant solvent physical properties, such as polarity and dielectric constant influencing dye solvation and/or shielding the intermolecular interaction, but also by the presence of third particles, i.e., colloidal organic or inorganic NPs, which provide a templating effect at the NP interface, leading to dye immobilization in the different aggregate forms [25,26,29,30]. For example, Mancin et al. [25] performed the aggregation engineering of a negatively charged porphyrin dye triggered by positively charged Au NPs, with the consequent change of the hybrid system photophysical properties. Byjdak [30] reviewed how layered NPs influence the molecular aggregation of organic dyes, by taking into account dye molecular structures, NP surface charge, solvent properties and ionic strength. Zu et al. described the templating effect played by the metal oxide NPs in dye aggregation towards an effective sensitization of metal oxide electrodes to obtain a high efficiency of dye-sensitized solar cells [29].

Functionalization of plasmonic NPs active in the NIR region by NIR absorbing dyes is a new field which, due to the synergistic effect for boosting infrared light harvesting by both dyes and NPs [31], can be targeted at specific applications, such as improving the performance of solar cells, NIR light upconversion [8], development of nano-heaters in targeted photothermal therapy [32,33], and conversion of heat into thermoacoustic waves for image reconstruction [34], where the above effects associated with absorption or release of thermal energy from/to the environment can be controlled by NP surface chemistry and corresponding surface functionalization. On the other hand, plasmonic NPs can be used to control emission of the immobilized dyes in the NIR, which is important, for example, in biophotonic applications [35].

In this work, NIR-absorbing plasmonic NPs in the form of copper sulphide NPs (with the general formula Cu_2-x_S) are used as a viable material [36]. These plasmonic NPs have some advantages over noble metals because they are less expensive and more stable upon exposure to light [37]. Because their LSPR band covers the second (II)-NIR window (i.e., from 1000 to 1700 nm), they have a potential for biomedical applications [38,39]. On the other hand, a dye of the tricarbocyanine family, whose monomer, dimer or aggregate absorption covers the first-I NIR window, is selected for combination with the above NPs. Because both materials have absorption in complementary NIR regions, the corresponding hybrid structure will potentially cover a wider spectral range and is expected to be more effective for applications where thermal energy harvesting is needed. In addition, if a hot-band absorption of the dye overlaps with the plasmonic band of the employed NPs, one can expect enhanced anti-Stokes emissions for energy upconversion purposes [40,41]. However, while the dye is soluble in organic polar solvents, the Cu_2-x_S NPs are only dispersible in apolar solvent. Therefore, the Cu_2-x_S NPs were coated with a mesoporous silica shell (Cu_2-x_S@MSS), which, besides improving the dispersibility issue, provides a high specific surface area [35] to enhance the dye–NP interaction, and also increases the dye–Cu_2-x_S NP distance through the MSS shell to prevent their direct contact and also to protect the plasmonic core from detrimental interaction with the environment.

Here the role of the MSS mediating the dye–NP interaction and influencing the dye aggregation on the surface of NP is investigated. By introducing amino groups at the MSS surface (Cu_2-x_S@MSS-NH_2_) that provide a positively charge interface, the yield of the hybrid dye–NP assembly formation can be enhanced. An in-depth spectroscopic characterization is carried out by comparing the dye assembling on the surfaces of Cu_2-x_S@MSS and Cu_2-x_S@MSS-NH_2_ in order to clarify how the dye aggregate arrangement is triggered by the NP’s surface charge density and dispersant solvent. Different organization of the dye molecules on the surface of Cu_2-x_S@MSS-based nanostructures tunes their specific functional properties, such as NIR absorption and photoluminescence, thus offering a fundamental understanding towards further applications of these hybrid NPs.

## 2. Experimental

Chemicals. The following chemicals were used for the Cu_2-x_S NPs synthesis: oleic acid (OA, 90%), oleylamine (Olam, 70%), copper(II) chloride (CuCl_2_, 90%), sulphur (S_8_, 99.99%), 1-octadecene (ODE, 90%), chloroform (CHCl_3_), ethanol (EtOH), tetrachloroethylene (TCE). For the MSS growth, cetyltrimethyl ammonium bromide (CTAB), NaOH, tetraethyl orthosilicate (TEOS 98%), 3-aminopropyl triethoxy silane (APS), ethylacetate and HCl (32% aqueous solution) were purchased from Sigma-Aldrich (Milan, Italy). Dimethylsulfoxide (DMSO), isopropylalcohol (IPA), acetonitrile (ACN) and tetrahydrofuran (THF) were obtained from Sigma-Aldrich.

NP and dye synthesis. Cu_2-x_S NPs coated with a hydrophilic MSS, referred to further as Cu_2-x_S@MSS, and those additionally functionalized by amino groups (Cu_2-x_S@MSS-NH_2_), were synthesized as follows. First, Cu_2-x_S NPs were obtained by a hot-injection method under nitrogen flux using a standard Schlenk line [36]. CuCl_2_ powder (67.2 mg) was poured in a three-necked flask adding 1.5 mL (2.5 mmol) of OA, 3.5 mL (5 mmol) of Olam and 7.5 mL (10 mmol) of ODE, as a high boiling point solvent, while S_8_ (16 mg) was dispersed in OA (1 mL, 1.5 mmol) and 1.5 mL of ODE in another flask. After a first stage under vacuum at 100 °C, the two flasks were put under nitrogen flow, and the temperature was increased to 180 °C, for the copper precursor, and 150 °C for the sulphur precursor. Then, 2.5 mL of sulphur precursor was injected into the copper precursor flask; the reaction mixture was left to react for 10 min at 180 °C and then the NPs were recovered by three steps of addition of EtOH and centrifugation, to be finally redispersed in TCE for further characterization, resulting in a concentration of 60 μM. For the growth of the MSS, 300 μL of the prepared NP solution was diluted with CHCl_3_ to 500 μL as a preliminary step, and an equimolar mixture of OA and Olam (0.03 mmol of each ligand) was then added; then, the colloidal solution was stirred for one hour, prior to collection by centrifugation and redispersion in 500 μL of CHCl_3_. An amount of 5 mL of an aqueous solution of CTAB 50 mM was then added, the emulsion was stirred for 30 min and finally the organic solvent was removed under vacuum, resulting in a clear colloidal solution. Then, 45 mL of Milli-Q water and 3 mL of ethyl acetate were added into the flask, which was then closed with a rubber seal and placed under nitrogen flow to remove air. Finally, 0.3 mL of TEOS and NaOH solution to provide a final concentration of 5 mM were added. The solution was stirred for 2 h at 40 °C and for 1 h at 60 °C, and then kept at 25 °C overnight. The Cu_2-x_S@MSS NPs were collected by centrifugation at 13,000 g/redispersion in EtOH three times and then dispersed in 2 mL of EtOH. For the synthesis of Cu_2-x_S@MSS-NH_2_, the amino groups were grafted by the addition of APS (70 μL) and an equal volume of NH_4_OH to the Cu_2-x_S@MSS samples diluted as 1:3 in EtOH in the presence of CTAB, 5 mM. The reaction was carried out under stirring at 25 °C overnight, and then the unreacted precursors were removed by centrifugation and redispersion steps in EtOH. To remove the CTAB from the mesoporous structures, both the Cu_2-x_S@MSS and the Cu_2-x_S@MSS-NH_2_ samples were resuspended in 20 mL of ethanol solution containing 1 mM of HCl and sonicated for 3 h. To recover the NPs, repeated cycles of centrifugation and redispersion in water were carried out. The final pellet was redispersed in 2 mL of Milli-Q water, resulting in a concentration of 15 mg/mL for Cu_2-x_S@MSS and 7 mg/mL for Cu_2-x_S@MSS-NH_2_, as measured by freeze-drying an aliquot of the sample_._ A ninhydrin assay, as describe in Ref. [42], was exploited to estimate the density of amino-groups per NP.

NIR dye of a tricarbocyanine family (Figure 1) was synthesized according to the procedures described in Ref. [43].

Sample preparation. To prepare hybrid NPs via assembly of dye molecules on the surface of Cu_2-x_S@MSS and Cu_2-x_S@MSS-NH_2_, the dye was dissolved either in organic solvents, i.e., DMSO, IPA, ACN, THF, or in distilled water. Then, 1 mL of the dye solution in a selected solvent was mixed with 0.01 mL of a dispersion of Cu_2-x_S@MSS or 0.02 mL of Cu_2-x_S@MSS-NH_2_, resulting in a dye concentration of 10^−5^ M, and 0.15 g/L (0.015% *w*/*v*) and 0.14 g/L (0.014% *w*/*v*) for the Cu_2-x_S@MSS and Cu_2-x_S@MSS-NH_2_, respectively, in the mixture. Preliminary experiments using lower amounts of Cu_2-x_S@MSS-based NPs (0.07 g/L) were also carried out. The colloidal stability of the hybrid structures was monitored as a function time.

Characterization techniques. A Cary 5000 spectrophotometer (Varian Agilent Technologies Italia, Milan, Italy) was used to record the UV-VIS-NIR absorption spectra of the Cu_2-x_S, Cu_2-x_S@MSS and Cu_2-x_S@MSS-NH_2_. The solutions were diluted for characterization using TCE or ethanol and water as dispersant solvents. A Zetasizer nano-ZSP (Malvern, Worcestershire, United Kingdom) was used for dynamic light scattering (DLS) and ζ-potential measurement of the samples by diluting the colloidal solution at 7 μg/mL.

A JEOL JEM1011 (JEOL Akishima, Tokyo, Japan) electronic microscope operating at 100 kV, furnished with a high-resolution CCD camera, was used for TEM measurements by depositing the suspension on a carbon-coated copper grid. Cu_2-x_S NCs colloidal solution diluted by 1:250 with TCE and 3 μL of the colloidal solution were drop-cast onto the TEM grid, allowing the solvent to evaporate. A similar procedure was used to deposit the Cu_2-x_S@MSS-based samples onto the TEM grid. The ImageJ analysis freeware was used for statistical analysis of the TEM images, measuring the average diameter of nearly 150 particles, and their size distribution was calculated as percentage relative standard deviation (σ%).

Absorption and photoluminescence (PL) spectra of the dye and hybrid NP dispersions were measured using an AvaSpec-2048 spectrophotometer. For these studies, solutions were prepared in 10 mm quartz cuvettes, with a pure solvent serving as a reference. Fluorescence was excited by a solid-state diode laser operating at wavelength of 780 nm (producing a power of 100 mW). The PL emission signal was collected by a 600 μm fibre at a right-angle in respect to the excitation light direction and registered by a CCD detector.

## 3. Results and Discussion

*Synthesis of Cu_2-x_S@MSS-based nanoparticles.* Cu_2-x_S@MSS NPs with an average size of 52 nm (σ% 8, Figure 1b) were obtained by the multistep approach [37], as described in the experimental section. Specifically, the first stage of synthesis in organic solvent provided Cu_2-x_S NPs of nearly 10 nm (σ% 15, Figure 1a), followed by ligand shell “reconstructuring” by OA and Olam treatment, phase transfer in water, where a lipophilic interaction of the CTAB surfactant with the alkyl chain of the ligands at the NP surface occurred, and finally in situ growth of a uniform MSS around Cu_2-x_S NP by sol–gel reaction of TEOS precursor under an alkaline environment. An average thickness of 21 nm (σ% 11) of the MSS was obtained by the precise control of the TEOS volume in the aqueous colloidal solution of the surface-modified Cu_2-x_S NPs (0.3 mL of TEOS per ~50 μM of NPs). The obtained final size of the NPs possessed several advantages as it did not allow high light scattering, and it also provided high specific surface area and protection of the Cu_2-x_S NPs against the environment. Even though functionalization with MSS brought a partial decrease in the plasmon band (Figure 1d, blue line) compared to the pristine Cu_2-x_S NPs (Figure 1d, dashed line), the characteristic absorption features ascribed to NP semiconducting and plasmonic characteristic covering the UV and the NIR region were still clearly visible in the spectra of Cu_2-x_S@MSS NPs (Figure 1d, blue line).

Functionalization of the Cu_2-x_S@MSS NPs with amino-groups was carried out according to previously reported procedure [42], which exploited the reactivity of the silanol groups at the MSS surface with siloxane moieties of APS, leading to amino-group grafting at the MSS surface (Figure 1c). The reaction was carried out in the presence of CTAB at a concentration higher than the critical micelle concentration to preserve the mesoporous structure. The absorption line profile of Cu_2-x_S@MSS-NH_2_ somewhat changed compared to Cu_2-x_S@MSS NPs (Figure 1d). A Ninhydrin assay was performed to qualitatively and quantitatively provide the success of the NP functionalization. It selectively reacted, in the presence of 2,6 lutidine, with primary amino groups. Because this reaction is accompanied by changing the solution colour to blue due to Ruhemanns Blue by-product formation, the effectiveness of the functionalization was spectroscopically confirmed by monitoring the absorption in the visible range. The quantity of amino groups per NP of nearly 10^5^ was estimated, after plotting a titration curve. The functionalization was also confirmed by the ζ-potential measurements of the samples after the treatment with dilute HCl solution (Figure 1e). While the Cu_2-x_S@MSS NPs exhibited a negative ζ-potential value of nearly −29.1 ± 0.9 mV, functionalization with APS (Cu_2-x_S@MSS-NH_2_) turned the ζ-potential value to +23.2 ± 1.4 mV. The ζ-potential values suggested that both colloidal solutions of Cu_2-x_S@MSS and Cu_2-x_S@MSS-NH_2_ possessed a good colloidal stability.

Behaviour of the NIR dye in different solvents. The NIR dye selected for this work belongs to the family of tricarbocyanine NIR dye salts bearing negatively charged sulphonate substituents at the nitrogen atoms in the heterocyclic core and a trialkylammonium counterion (see Figure 1). Dispersion of this dye in polar organic solvents usually results in complete dissolution of the dye powder into monomers, whereas a poor solvent can result in aggregation phenomena induced by dye–dye van der Waals, electrostatic, and hydrophobic interactions. Figure 2 shows the UV-Vis-NIR absorption spectra of the as-prepared dye solutions in the different solvents used in this work.

In organic polar solvents of DMSO, ACN, THF and IPA, the dye monomer indicated an absorption maximum around 840 nm, followed by a vibronic sideband absorption tail near ~760 nm; however, the spectral features also followed a solvatochromic shift, which is roughly dependent on the solvent polarity [44], indicating good dye–solvent interaction. In contrast, the dye molecules experience hydrophobic interactions in water, leading to a high yield of dimers with absorption maximum at ~750 nm, followed by a subsequent formation of J-aggregates with a characteristic red-shifted absorption band at ~940 nm (Figure 2).

Formation of hybrid nanoparticles in organic solvents. In order to prepare hybrid NPs, the dye solution was added to a colloidal dispersion of NPs, either Cu_2-x_S@MSS or Cu_2-x_S@MSS-NH_2_ NPs. The NP concentration in the mixture was set at 0.15 mg/mL and 0.14 mg/mL for Cu_2-x_S@MSS and Cu_2-x_S@MSS-NH_2_, respectively, to reduce scattering phenomena, meanwhile ensuring the dye–NP interaction to form hybrid nanostructures, which was easily controlled by UV-Vis-NIR absorption (see Appendix A). Because the photophysical properties of the dye molecules changed as these assembled onto the NP surface, absorption spectroscopy represented an effective tool useful for monitoring the hybrid NP formation. The physical adsorption of the dye onto the NPs was found to depend on the physical properties of the solvent used, which mediates the strength of the dye–NP interaction. In addition, the mesoporous shell is expected to enhance the adsorption process due to the porous structure and high specific surface area of the MSS (Figure 1c).

The effectiveness of the dye–NP interaction and the way by which the dye is assembled onto the mesoporous silica surface can be described based on the monitoring of both the photophysical properties and the colloidal stability of the mixtures. In Figure 3, the UV-Vis-NIR absorption spectra of the mixture containing dye and Cu_2-x_S@MSS-NH_2_ (Figure 3a) and Cu_2-x_S@MSS (Figure 3b) NPs in ACN are shown as an example. The absorption spectra of the samples in DMSO, THF and IPA are reported in the Appendix A.

First, the absorption spectra of the dye and NPs mixtures (Figure 3, magenta curves) did not represent a sum of absorption of the separate components (Figure 3 a,b). In the presence of Cu_2-x_S@MSS-NH_2_ and Cu_2-x_S@MSS in ACN, the dye absorption maximum at ~833 nm become reduced and shifted to 837 nm and 835 nm, respectively (see Appendix A), and its bandwidth broadened, mostly developing from the red wing, which is characteristic of the dye–NP interaction [45]. However, these changes were found to be more pronounced for the dye in the presence of Cu_2-x_S@MSS-NH2 compared to Cu_2-x_S@MSS, which indicated better interaction of the dye sulfonate groups with grafted amino-groups at the NP surface compared to the bare MSS. Indeed, clear precipitation of the dye:Cu_2-x_S@MSS-NH_2_ hybrid could be observed within several hours (see inset in Figure 3a), whereas only a small amount of pure Cu_2-x_S@MSS precipitate could be observed during the same time period, with no sign of dye involved in the precipitate in the dye–Cu_2-x_S@MSS mixture (inset in Figure 3b). This suggests a strong interaction occurring between Cu_2-x_S@MSS-NH_2_ and the dye, which is accompanied by the reduction of colloidal stability for this hybrid nanostructure.

Table 1 indicates the absorption wavelength and absorption intensity of the monomers and a blue-shifted shoulder near 750 nm (originating due to superposition of the vibronic sideband and dimer absorption), and their relative absorption ratio as well, in different solvents with and without the Cu_2-x_S@MSS-NH_2_ NPs. A clear bathochromic shift of the monomer band and the blue-shifted shoulder is observed in the presence of Cu_2-x_S@MSS-NH_2_ using ACN as a dispersant solvent. Moreover, the relative mon/dim absorption ratio decreased for all colloidal solutions (Table 1), thus suggesting a minor formation of dimers on the surface of the Cu_2-x_S@MSS-NH_2_ NPs. The observed broadening of the absorption band for the dye:Cu_2-x_S@MSS-NH_2_ hybrid from the blue wing confirmed the formation of some dimers on the surface of NPs, whereas the dye persisted in its monomer form on the Cu_2-x_S@MSS surface (Figure 3b and Appendix A).

Signs of the dye assembly could also be seen from the changes in PL spectra. Compared to the neat dye, the quantum yield (QY) of PL emission decreased for the dye:Cu_2-x_S@MSS hybrid by a factor of ~2 and for the dye:Cu_2-x_S@MSS-NH_2_ hybrid by a factor of ~40 (Figure 4). The latter can potentially be due to the dye aggregation and/or plasmonic quenching effect due to the free carriers of Cu_2-x_S NPs. Because the aggregated form of the dye on the surface of Cu_2-x_S@MSS-NH_2_ was insignificant in the organic solvents, the plasmonic effect should be considered as the major reason for the PL quenching. Here, the distance between the plasmonic core and the immobilized dye was determined by the silica shell thickness (~20 nm), which most probably was not optimal for enhancement of the PL emission; in addition, the dye absorption wavelength (840 nm) was relatively far from the plasmonic maximum of Cu_2-x_S@MSS-NH_2_ at ~1100 nm. Usually, the fluorophores that emit in the UV-Vis are quenched in the case of a direct contact or when being approached to the plasmonic NP surface closely (i.e., within ~5 nm) [46]. However, in the NIR region, where the wavelength increases significantly compared to the UV-Vis, NIR plasmonics can require even larger dye–NP separation to escape PL quenching of the fluorophore [47].

At the same time, plasmonic absorption of the NPs rendered a clear effect on the hot-band absorption (HBA) of the dye (see inserts in Figure 3). HBA originates from absorption of thermally excited electrons being on the vibronic levels of the ground state (Figure 2) and can be observed as a small feature near 940 nm in the low-energy domain relative to the main absorption band in the NIR spectra (Figure 3). Spectrally, the HBA locates much closer to the plasmonic absorption maximum at ~1100 nm and overlaps with this broad plasmonic absorption band. As a result, in the presence of plasmonic NPs, the HBA of the dye increased (See inserts in Figure 3), which can be suggested to be either due to the plasmon resonance enhanced electric field or due to the increasing population of the hot band of the dye because of the thermal heating effect from the plasmonic NPs. Interestingly, the HBA increase was also dependent on the solvent used (See HBA for THF, DMSO and IPA solutions in the Appendix A). Because the solvents have different specific heat capacities (Appendix A), the explanation of the HBA enhancement via the increasing population of the hot band of the dye looks more feasible. Specifically, increasing specific heat capacity of the environment affects HBA-induced anti-Stokes emission via the decrease in the hot band population [44]. Among the solvents used in this work, IPA possesses the highest specific heat capacity (2.68 J/g·K) and resulted in the least HBA changes (Appendix A), whereas THF, DMSO and ACN, which have specific heat capacities within ~1.8 to 2.2 J/g·K, led to a significant increase in the HBA.

According to the observed spectroscopic features and temporal colloidal stability behaviour of the mixtures, a distinct mechanism of the dye assembly on the surfaces of Cu_2-x_S@MSS-NH_2_ and Cu_2-x_S@MSS can be derived, based upon the different surface chemistries and charge densities of the NPs, as well as the dye structure. As expected, Cu_2-x_S@MSS-NH_2_ can bind dye molecules through direct interaction of the positively charged amino-groups (ζ-potential value + 23.2 mV) of the functionalized NP’s shell and negatively charges sulfonate-groups of the dye molecule. Because both the sulfonate-groups of the dye and the amino-groups of the NPs are now screened from the solvent, the new interface of the hybrid nanostructures is expected to be poorly polar; hence, the hybrid NPs partially loss their colloidal stability in the polar organic solvent, as evidenced by the observed tendency to precipitate with time, compared to the single separate components (dye and NPs), which are stable in the selected solvent. Conversely, the surface of the Cu_2-x_S@MSS NPs is charged negatively, as highlighted by the measured value of the ζ-potential, being −29.1 mV; therefore, other types of interaction rather than electrostatic should drive the formation of this hybrid structure. Due to the presence of the positively charged dye salt counterions, which can easily interact with the negatively charged mesoporous silica shell first [48], the particle surface becomes more neutral and less resistant to adsorption of the dye chromophore. The latter can be assembled on the silica shell via interaction of the dye sulfonate groups and silanol groups normally present on the silica surface [49].

As highlighted in Table 1 and reported in Appendix A, the templating effect due to Cu_2-x_S@MSS-NH_2_ NPs could be changed by variation of the solvent medium. Indeed, the dispersant solvent tunes the strength of the dye–NP interaction through the dielectric constant and solvent polarity. The four polar organic solvents selected in this work showed the following trends of dielectric constant: DMSO (47) > ACN (37) > THF, IPA (17) (See Appendix A), with IPA being a sole protic solvent. The higher dielectric constant is expected to better shield the dye–NP electrostatic interaction, which drives the assembly of dye onto the Cu_2-x_S@MSS-NH_2_NPs surface. Indeed, the absorbance features were reduced to a higher extent for THF > ACN > DMSO (See Table 1), and concomitantly a larger decrease in the mon/dim absorption ratio was observed according to the same solvent sequence, suggesting that the dye mainly interacts with the NPs by electrostatic attraction. The sole exception was set by IPA, which, being a protic solvent, can strongly interact with both dye and NPs, thus effectively limiting their mutual interaction.

Formation of hybrid nanoparticles in aqueous solutions. The dye molecules used in this study are insoluble in water in the monomer form. Instead, they aggregate as H-type dimers (stacking structure) or J-aggregates (head-to-tail slippery arrangement of the neighbouring molecules). Specifically, dimers are often the building blocks for J-aggregates of merocyanine dyes [50], as was also observed here. Upon addition of a small amount of the DMSO solution of the dye monomers to water, immediate dimerization of the molecules occurred, which was displayed as the absorption band of the dimers at ~760 nm (Figure 5). However, over minutes this band gradually reduced and disappeared, along with the rise of the J-band absorption at ~940 nm. That is, the dimers united and thus gave rise to J-aggregation.

However, the presence of Cu_2-x_S@MSS-NH_2_ or Cu_2-x_S@MSS NPs in the aqueous dispersion changed the above dye aggregation dynamics substantially. In particular, when the dye was added to the Cu_2-x_S@MSS dispersion, the presence of these NPs led to disentanglement of the formed J-aggregates, which was evidenced by the modest increase in the monomer band in the absorption spectra, while the presence of dimers was highly suppressed (Figure 6a).

This behaviour can be interpreted as dye assembly in the monomer form on the surface of NPs at the expense of decay of J-aggregates. That is, the dye–Cu_2-x_S@MSS interaction overcomes the dye–dye interaction. Assembly of the dye in the monomer form on the Cu_2-x_S@MSS surface was also confirmed by PL emission spectrum, which showed the typical features of the dye monomer, albeit strongly suppressed due to the plasmonic quenching (Figure 4b). In the presence of Cu_2-x_S@MSS-NH_2_ NPs, however, J-aggregation was completely suppressed from the very beginning, and dye molecules assembled on the amino-functionalized MSS predominantly in the form of dimers (Figure 6b). In the case of the dye:Cu_2-x_S@MSS-NH_2_ hybrid NPs, dye emission was almost completely quenched, as dye dimers are non-emissive (Figure 4b).

It is interesting that formation of the hybrids could be traced not only via the addition of dye solution to the NPs dispersion, but also via the addition of NPs to the aqueous dye solution, which yielded somewhat different results. In this case, J-aggregates in aqueous dye solution were formed first, with a predominant absorption band at ~940 nm (Figure 7). Upon the addition of Cu_2-x_S@MSS NPs, J-aggregates were gradually destroyed, while the dye monomer amount increased (Figure 7a). Upon the addition of Cu_2-x_S@MSS-NH_2_ NPs, the J-aggregates were also gradually destroyed, but they transformed to dimers rather than to monomers (Figure 7b). Therefore, specific surface chemistry/charge density at NP surfaces drives the dye assembling differently.

## 4. Conclusions

We demonstrated functionalization of silica-coated NIR plasmonic Cu_2-x_SNPs by NIR dye, where the specific surface chemistry of the NPs drives dye assembling on the NP’s surface differently. In the case of the Cu_2-x_S@MSS-NH_2_ NPs, the driving force for the dye–NP interaction is the Coulomb interaction of sulfonic terminal groups of the dye molecule and amino-groups at the NP surface. This driving force is so high that it strongly competes with assembling J-aggregate from dye dimers in the aqueous medium, effectively preventing this process and accepting all dye dimers onto the NP surface. When the dye J-aggregates form first, the above driving force disassembles them into dimers as a result of dye immobilization on the NP surface. In the case of the Cu_2-x_S@MSS NPs, the driving force for the dye assembling on NPs most probably is due to the interaction of the sulfonic terminal groups of the dye molecule and hydroxyl-groups usually formed on the silica surface. This driving force is weaker compared to the previous case. It destroys dye J-aggregates gradually, converting them predominantly into monomers, as a result of the dye–silica shell interaction. In addition, it was shown that the driving force for the dye assembling is strongly dependent on the solvent used. In polar solvents, the driving force decreases with the increasing dielectric constant of the solvent, promoting the shielding of the dye–NP electrostatic interaction. Thus, depending on the surface chemistry of NPs and the solvent environment, different scenarios of dye aggregation on the NP’s surface can be obtained. This provides further potential for application issues of these dye–NP hybrids, such as light harvesting and PL emission in the near-infrared.

## Data Availability

The data presented in this study are reported in the article and in the Appendix A. Additional details can be requested from the corresponding authors.

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
