# Peer review of "Assembling Near-Infrared Dye on the Surface of Near-Infrared Silica-Coated Copper Sulphide Plasmonic Nanoparticles"

_nanomaterials, 2023, doi:10.3390/nano13030510_

Round 1

Reviewer 1 Report

In their work entitled “Assembling Near-Infrared Dye on the Surface of Near-Infrared Silica-Coated Copper Sulphide Plasmonic Nanoparticles” Dimitriev and co-workers show functionalization of plasmonic nanoparticles resonating in the II NIR window with tricarbocyanine NIR dye resonating in the I NIR window and possessing sulfonate end groups. In more detail, they show the role of specific surface chemistry in dye assembling on the surface of the nanoparticles through the use of either organic polar liquids or water used as a dispersant solvent. In the first case it is shown that dye binding to the nanoparticles surface occurs with different efficiency but mostly in the monomer form. In the second case, that is in water, they observe different scenarios depending on the NP surface chemistry. They observe predominant formation of the disordered dye monomers on the bare surface of mesoporous silica shell, whereas the amino-group functionalized silica shell accepts dye predominantly in the form of dimers. Interestingly, it is found that the dye-nanoparticle interaction overcomes the dye-dye interaction. The article is well written and organized, and the study itself represents a nice investigation of surface chemistry in nanoscale systems with potential applications in medicine and light harvesting. Overall, I think the article is worth to be published in Nanomaterials, although I have some minor points I ask the authors to consider before giving my final opinion.

1. It is not clear what is the advantage to use a system whose components (in this case dye and nanoparticle) work in different spectral ranges. To me, it would make more sense if the plasmonic properties are used to enhance or modify the chemical properties of the dye, for instance for enhanced spectroscopy (see for instance Scientific Reports 2018, 8, 12652 or Nature Communications 2019, 10, 5321).

2. Why this dye has been chosen? The authors should motivate this better, as it is not clear to what extent this system can be used for light harvesting and/or in other fields. The authors should provide a broader overview of the applications of the system.

3. Although the nanoparticles resonate in the II NIR window, how can the author exclude thermal effects upon light excitation of resonance?

4. Interestingly, the different organization of the dye molecules on the surface of silica-coated copper sulphide nanoparticles might provide a knob to tune their specific functional properties. I can see for photoluminescence, but how this would impact light harvesting applications.

5. I suggest, for instance in Figure 3, to put an arrow to show what is the monomer and dimer peak.

6. Finally, if plasmonic properties are not used, why such an emphasis on the fact that the nanoparticles used have plasmonic properties (also in the title)? From this, it looks like plasmons are playing a role, while the whole paper is mainly devoted to study surface chemistry effects. What is then the role of the nanoparticles?

Author Response

First, we thank the reviewer for the constructive critique which is beleived to make the revised manuscript better. Point by point answers to the comments are given below. The corresponding changes along with the language polishing are indicated in the revised manuscript by yellow.

  1. It is not clear what is the advantage to use a system whose components (in this case dye and nanoparticle) work in different spectral ranges. To me, it would make more sense if the plasmonic properties are used to enhance or modify the chemical properties of the dye, for instance for enhanced spectroscopy (see for instance Scientific Reports 2018, 8, 12652 or Nature Communications 2019, 10, 5321).

Answer: Thank you for this comment. We added suggested references as indicated specific application of plasmonic modified NPs. In turn, we also added the evidence of the increasing hot-band absorption of dye molecules as a “neat” plasmonic effect due to  interaction with plasmonic NPs.

  1. Why this dye has been chosen? The authors should motivate this better, as it is not clear to what extent this system can be used for light harvesting and/or in other fields. The authors should provide a broader overview of the applications of the system.

Answer: First, the dye was chosen due to specific sulfonic groups able to interact with the selected NPs in order to show the different scenarios of assembling. Second, the selected dye possesses hot-band absorption (HBA) which spectrally overlaps with plasmonic absorption band of NPs, which leads to the dye HBA enhancement. Although this topic was planned as a separate research work, we added more information concerning applicability and effect of plasmonic NP on the dye spectral properties.

In introduction: “Also, when hot-band absorption of the dye overlaps with the plasmonic band of the employed NPs one can expect enhanced anti-Stokes emission for energy upconversion purposes [40,41].”

In Results and Discussion: “At the same time, plasmonic absorption of the NPs renders a clear effect on the hot-band absorption (HBA) of the dye (See inserts in Fig. 3). HBA originates from absorption of thermally excited electrons being on the vibronic levels of the ground state (Scheme 2), and can be observed as a small feature near 940 nm in the low-energy domain relatively to the main absorption band in the NIR spectra (Fig. 3). Spectrally, the HBA locates much closer to the plasmonic absorption maximum at ~1100 nm and overlaps with the broad plasmonic absorption band. As a result, in the presence of plasmonic NPs, the HBA of the dye increases (See inserts in Fig. 3), which can be suggested to be either due to the plasmon resonance enhanced electric field or due to increasing population of the hot band of the dye because of the thermal heating effect from the plasmonic NPs. Interestingly, the HBA increase is dependent on the solvent used too (See HBA for THF, DMSO and IPA solutions in Supplementary Materials, Fig. S2). Since the solvents have different specific heat capacities (Table S2), the explanation of the HBA enhancement via increasing population of the hot band of the dye looks more feasible. Specifically, increasing specific heat capacity of the environment affects HBA-induced anti-Stokes emission via the decrease of the hot band population [43]. Among solvents used in this work, IPA possesses highest specific heat capacity (2.68 J/g·K) and results in the least HBA changes (Fig. S2), whereas THF, DMSO and ACN which have specific heat capacities within ~1.8 to 2.2 J/g·K, lead to significant increase in the HBA. “

  1. Although the nanoparticles resonate in the II NIR window, how can the author exclude thermal effects upon light excitation of resonance?

Answer: No, the thermal effect cannot be excluded, but it has a useful effect here as it enhances dye HBA (see above comment).

  1. Interestingly, the different organization of the dye molecules on the surface of silica-coated copper sulphide nanoparticles might provide a knob to tune their specific functional properties. I can see for photoluminescence, but how this would impact light harvesting applications.

Answer: Again, we added the results on enhancement of dye HBA  (see comment #2), which means that light harvesting in the NIR becomes improved.

  1. I suggest, for instance in Figure 3, to put an arrow to show what is the monomer and dimer peak.

Answer: Strictly speaking, in Fig. 3, only the monomer band can be determined clearly, whereas the blue-shifted shoulder corresponds to superposition of the dimer and sideband vibronic absorption. Therefore, we refer to Fig. 2, where the bands of the monomer, dimer, and J-aggregates are determined unambiguously.

  1. Finally, if plasmonic properties are not used, why such an emphasis on the fact that the nanoparticles used have plasmonic properties (also in the title)? From this, it looks like plasmons are playing a role, while the whole paper is mainly devoted to study surface chemistry effects. What is then the role of the nanoparticles?

Answer: Again, we added the results on enhancement of dye HBA  (see comment #2), which is a direct consequence of the plasmonic effect.

Reviewer 2 Report

The authors demonstrated functionalization of mesoporous silica-coated copper sulphide NPs with NIR dyes. The authors explored effects of aqueous medium, amino-group on the dye-NP hybrid formation. They concluded that the driving force for the dye assembling on Cu2-xS@MSN-NH2 disassembles dye J-aggregates into dimers. While the dye-Cu2-xS@MSS NPs interactions convert dye J-aggregates predominantly into monomers. The authors also demonstrated photoluminescence of dye-NP hybrid in aqueous dispersions is stronger than j-aggregates dye in aqueous dispersions.

This is nice work on adjustable functionality of the dye-NP hybrid. However, I have several concerns.

1. Although the silica-coated copper sulphide NPs provide tunable functionalities, but it does not provide any enhancement to the absorbance. Is it possible to utilize the plasmonic nature of NPs to enhance the absorption?

2. Why the PL quantum yield for dye-NP hybrid in aqueous dispersion is much smaller than the hybrid in ACN?

3. The supplementary material is missing in the submission. Please submit them.

Author Response

First, we thank the reviewer for the constructive critique which is beleived to make the revised manuscript better. Point by point answers to the comments are given below. The corresponding changes along with the language polishing are indicated in the revised manuscript by yellow.

  1. Although the silica-coated copper sulphide NPs provide tunable functionalities, but it does not provide any enhancement to the absorbance. Is it possible to utilize the plasmonic nature of NPs to enhance the absorption?

Answer: Yes, we show that the hot-band absorption of the dye can be enhanced. We modified Fig. 2 and added Fig. S2 along with explanation in the text as follows: “At the same time, plasmonic absorption of the NPs renders a clear effect on the hot-band absorption (HBA) of the dye (See inserts in Fig. 3). HBA originates from absorption of thermally excited electrons being on the vibronic levels of the ground state (Scheme 2), and can be observed as a small feature near 940 nm in the low-energy domain relatively to the main absorption band in the NIR spectra (Fig. 3). Spectrally, the HBA locates much closer to the plasmonic absorption maximum at ~1100 nm and overlaps with the broad plasmonic absorption band. As a result, in the presence of plasmonic NPs, the HBA of the dye increases (See inserts in Fig. 3), which can be suggested to be either due to the plasmon resonance enhanced electric field or due to increasing population of the hot band of the dye because of the thermal heating effect from the plasmonic NPs. Interestingly, the HBA increase is dependent on the solvent used too (See HBA for THF, DMSO and IPA solutions in Supplementary Materials, Fig. S2). Since the solvents have different specific heat capacities (Table S2), the explanation of the HBA enhancement via increasing population of the hot band of the dye looks more feasible. Specifically, increasing specific heat capacity of the environment affects HBA-induced anti-Stokes emission via the decrease of the hot band population [43]. Among solvents used in this work, IPA possesses highest specific heat capacity (2.68 J/g·K) and results in the least HBA changes (Fig. S2), whereas THF, DMSO and ACN which have specific heat capacities within ~1.8 to 2.2 J/g·K, lead to significant increase in the HBA. “

  1. Why the PL quantum yield for dye-NP hybrid in aqueous dispersion is much smaller than the hybrid in ACN?

Answer: The PL spectra in Fig. 4 represent superposition of PL emission both from dye monomers immobilized on the surface of NPs and from free monomers in the solution which did not react with the NPs. Since in aqueous solution no free monomers are present and the PL intensity comes only from immobilized molecules (aggregates have very low QY of emission if any), therefore, its intensity is much smaller than in case of ACN, where free monomers are available.

  1. The supplementary material is missing in the submission. Please submit them.

Answer: sorry for that, supplementary material is added to the submission.

Round 2

Reviewer 2 Report

The authors have addressed all my concerns.